# Antimicrobial Activity of a 3D-Printed Polymethylmethacrylate Dental Resin Enhanced with Graphene

**DOI:** 10.3390/biomedicines10102607

**Published:** 2022-10-17

**Authors:** Helena Salgado, Ana T. P. C. Gomes, Ana S. Duarte, José M. F. Ferreira, Carlos Fernandes, Maria Helena Figueiral, Pedro Mesquita

**Affiliations:** 1Faculty of Dental Medicine (FMD), Universidade Católica Portuguesa, 3504-505 Viseu, Portugal; 2Centre of Interdisciplinary Research in Health, Faculty of Dental Medicine (FMD), Universidade Católica Portuguesa, 3504-505 Viseu, Portugal; 3Department of Materials and Ceramic Engineering, CICECO, University of Aveiro, 3810-193 Aveiro, Portugal; 4Faculty of Engeneering (FEUP), Universidade do Porto, 4200-465 Porto, Portugal; 5Faculty of Dental Medicine, Universidade do Porto, 4200-393 Porto, Portugal

**Keywords:** polymethyl methacrylate, graphene, *Candida albicans*, *Streptococcus mutans*, denture stomatitis, 3D printing

## Abstract

The present study aimed to test, in vitro, the antimicrobial activity against *Candida albicans* and *Streptococcus mutans* and the surface roughness of a 3D-printed polymethylmethacrylate dental resin enhanced with graphene. A 3D-printed polymethylmethacrylate dental resin was reinforced with four different concentrations of graphene: 0.01, 0.1, 0.25 and 0.5 wt%. Neat resin was used as a control. The specimens were printed in a liquid crystal display printer. Disc specimens were used in antimicrobial evaluation, and bar-shaped specimens were used to measure surface roughness. The study of antimicrobial activity included the inhibition of the growth of *C. albicans* and *S. mutans* and their adhesion to the resin’s surface. Surface roughness increased with the increase in the graphene concentration. The growth inhibition of *C. albicans* was observed in the different concentrations of graphene after 24 h, with no recovery after 48 h. The specimens doped with graphene were capable of inactivating *S. mutans* after 48 h. The surface-adhesion studies showed that the density of microbial biofilms decreases in the case of specimens doped with graphene. Graphene, despite increasing the resin’s surface roughness, was effective in inhibiting the growth and the adhesion to the resin’s surface of the main inducers of prosthetic stomatitis.

## 1. Introduction

Up to the middle of the 20th century, the great majority of the elderly was edentulous. However, during the last few decades, due to an improvement in dental care combined with a professional oral care approach that follows the principles of prevention and minimally invasive dentistry, a reduction in the prevalence of edentulism and the incidence of tooth loss has occurred in many industrialized countries [1]. 

Therefore, despite the improvement in oral health care, the increase in life expectancy leads to prosthetic rehabilitation continuing to be a need with high prevalence. The World Health Organization (WHO) predicts that in 2050, the percentage of the population that is over 60 years of age will be close to 30% in developed countries [2].

Polymethylmethacrylate (PMMA) is still a clinician’s first choice for various prothesis fabrications: denture bases, fabrication of artificial teeth, temporary crowns, obturators for cleft palates and denture relining and repair. A drawback of conventional denture acrylic resins is the high susceptibility to microbial colonization in the oral environment due to some properties such as porosity, surface charge, amount of surface free radicals, hydrophobicity and surface roughness [3,4]. So, undoubtedly, prosthetic devices themselves play a critical role in the colonization of bacteria and yeast and the formation of biofilm, which determines the development of some oral diseases [5,6].

Some studies [7,8] have demonstrated that surface roughness significantly influenced the extent of microbial adhesion to the denture base. Microbial attachment was increased on rougher surfaces.

The adhesion of microorganisms to hard dental surfaces is followed by the formation of plaque [9]. Surface roughness plays an important role in this process [9,10]. Changes in this physical property have clinical consequences on the changes in bacterial adhesion. 

Denture stomatitis is a very common disease that affects denture users. It has a high prevalence and may affect, according to some studies, two-thirds of removable denture wearers [11,12,13]. Although it is a common disorder, its etiology is still little understood and is considered to be multifactorial [14]. Some factors have been considered to be predisposing of this pathology such as systemic conditions (immunocompromised patients), increasing age of the denture users, mucosal trauma due to poor denture fit, bacterial and fungal infection, increased age of dentures and poor denture hygiene [15]. 

*Candida albicans* is an opportunist fungus found in human mucosa, but, under local or systemic conditions, it might invade tissues and become harmful. It is responsible for denture stomatitis in 50% to 90% of infections, but other species are also present, such as *C. tropicallis* or *C. glabrata* [16,17]. 

Studies have shown that denture stomatitis is not only a result of *Candida spp.* but of multispecies biofilms that may include *Streptococcus mutans* and *Staphylococcus aureus. S. mutans* is often found in the surface of acrylic resins and when incubated together with *C. albicans,* can compete for biding sites, but it also may favor yeast adhesion [18].

The increasing use of antifungal drugs, such as nystatin, fluconazole, itraconazole or chlorhexidine, in the treatment of denture stomatitis has shown some serious side effects and, additionally, has led to the development of resistant fungal and bacterial strains [19,20]. 

To overcome these problems, recent research has been carried out by incorporating nano additives, in the form of nanospheres, nanosheets, nanofibers or nanotubes, to the denture base resin. Recent studies have proven antifungal efficacy of various nano materials, such as zirconium, magnesium oxide (MgO), zinc oxide (ZnO), titanium oxide (TiO_2_) and silver nanoparticles, when mixed with polymers. Nanoparticles have a higher level of fungicidal effects than conventional antifungal medicines as they have better penetration in host cells and tissues, even in small concentrations [21].

Recent studies showed that, besides their remarkable mechanical properties, graphene-based materials have demonstrated antimicrobial effects against a wide range of bacteria and fungi [22,23], including an oral biofilm [24].

Graphene is a two-dimensional carbon nanomaterial consisting of a layer of single-atom-thick sp-2-bonded carbon atoms. It has a closely-packed hexagonal honeycomb structure that contains functional groups, such as epoxy, hydroxyl, carbonyl and carboxylic, on the surface and at the edges [25].

The antimicrobial effect of graphene and its derivates, graphene oxide (GO) and graphene oxide reduced (rGO), is very promising. There is evidence that graphene has the ability to reduce the viability of microorganisms in the oral cavity. The mechanism of antimicrobial action occurs by damage in the cell wall followed by cell lysis [24]. 

CAD/CAM technology is an increasingly evident reality in Dentistry. 3D printing, or additive manufacturing, is a technology that allows a quick planning and printing of 3D objects, so it is expected to be of great value in the prosthetic rehabilitation clinical practice. This manufacturing method involves adding layers of resin material incrementally, with light activation, to build the prothesis in the process. The new 3D printing technology leads to the development of new biomaterials such as PMMA photocurable resins whose properties need to be improved for biomedical applications [26,27]. A way that has currently been seen as promising in improving the mechanical and biological properties of these photocurable resins may be by incorporating nanomaterials into the resin. In a very recent study [28], it has been shown that graphene can be used as an additive of photocurable resins; however, these nanocomposites still need to be further studied, namely, mechanically to ensure that 3D-printed structures have these properties improved as well as biologically to verify the effect of graphene on microbial proliferation and adhesion.

The aim of this study was to evaluate the effect of graphene nanoplatelets added to a 3D-printed PMMA resin on surface roughness and antimicrobial activity against *Candida albicans* and *Streptococcus mutans*. The null hypothesis stated that there was no significant difference in the surface roughness and antimicrobial effect of PMMA with and without graphene incorporation.

## 2. Materials and Methods

### 2.1. Graphene–PMMA Resin Preparation

Graphene nanoplatelets (GNP) (Graphenest Advanced Nanotechonoly, Sever do Vouga, Aveiro Portugal) with 8–30 layers, a thickness of 3–10 nm and layers’ lateral dimensions of 0.5–2 µm were added to Dental Sans resin (HARZ Labs Inc., Moscow, Russia), which is the commercial name for a photosensitive liquid mixture of methacrylate oligomers and monomers. The graphene concentrations were: 0.01 wt%, 0.1 wt%, 0.25 wt% and 0.5 wt%. The resultant mixture was homogenously mixed to ensure the full dispersion of the nanomaterial, with a magnetic stirrer for 45 min at a setting of 1500 rpm. The viscosity of the resins, reinforced with the different graphene concentrations, was tested, and this property did not suffer significant changes with the presence of graphene, remaining at the value of 1.2 Pa.s. 

### 2.2. Design and Fabrication

To characterize the antimicrobial activity of the graphene–PMMA resin, a total of 70 cylinder-shaped specimens (5 mm in diameter, 1 mm in height) were used and divided into 5 groups: control group (Gc) and 4 experimental groups according to the different concentrations of graphene (G0.01, G0.1, G0.25 and G0.5). A total of 25 bar-shaped specimens (50 × 10 × 4 mm) were used to measure surface roughness, 5 for each group.

A CAD software Solidworks (Dassault Systems S.A, Vélizy-Villacoublay, France) was used to virtually design the specimens according to the above-mentioned dimensions (Figure 1). The CAD standard tessellation language (STL) files were sent to the 3D printer. 

The specimens were 3D-printed using a liquid crystal display (LCD) technology (Phrozen Mini 4K, Prozen, Hsinchu, Taiwan), which is equipped with a 405 nm light-emitting diode (LED) as the light source to cure the liquid polymer resin layer by layer. The fabrication process is illustrated in Figure 2. 

The printing direction was 45 degrees. The layer resolution was 0.05 mm for all samples, and the exposure time was 1.9 s. The density of the printed material was 1.20 g/cm^3^, and the viscosity was 1.2 Pa.s. The addition of graphene did not appreciably change these properties of the polymer.

After printing, all the specimens were cleaned using a 90% isopropyl alcohol bath in a postprocessing machine (Creality 3D UW-02, Creality 3D, Shenzhen, China) for 5 min to remove the excess liquid resin according to the manufacturer’s indications. Then the specimens were placed in an oven at 80 °C for 30 min (POL-EKO Aparatura, Wodzisław Śląski, Poland) for post-print baking. Postpolymerization was carried out in a UV light curing unit for 10 min (Creality 3D UW-02, Creality 3D, Shenzhen, China). The support structures were removed after post curing using low-speed rotary instruments (5000 rpm) (Figure 3).

The specimens were then finished, polished and measured by using a digital caliper with a 0.01 mm resolution (ABSOLUTE Digimatic Caliper Series 551, Mitutoyo Europe GmbH, Neuss, Germany), ensuring that the specimens conformed to the dimensional tolerances of the ISO 20795-1. After measurements, all specimens were disinfected by using 70% ethanol for 5 min and were stored in 37 °C water for 24 h according to ISO 20795-1.

### 2.3. Characterization of Graphene–PMMA Resin

Raman spectroscopy was carried out to identify the presence of GNP in the polymer resin at different concentrations. Measurements were carried out on an ALPHA300 R Confocal Raman Microscope (WITec, Ulm, Germany) using 532 nm laser light for excitation at room temperature (power ~1 mW to minimize localized heating and damage to the sample). The laser beam was focused on the sample by a 100× lens. Single acquisitions (5 s, 7 scans) were performed using the 600 g/mm grating. 

### 2.4. Surface Roughness Determination

The surface roughness profile for each specimen was measured using a contact profilometer (HommelWerke LV-50, Hommelwerke GMBH, Mannheim, Germany). The surface of the specimen was covered by a 5 μm ray diamond tip with constant load following a rectilinear measurement length of 4.8 mm for 10 s. Three measurements were performed for each specimen. The average roughness (Ra) was analyzed. Ra (µm) is the arithmetic mean value of all heights (peaks and valleys) in the given roughness profile.

### 2.5. Antimicrobial Activity of Graphene–PMMA Resin against C. albicans and S. mutans

#### 2.5.1. Microbial Strains and Growth Conditions 

In this study, *Candida albicans* ATCC 11225, stored at −80 °C in Sabouraud Dextrose broth (SDB, Liofilchem, Roseto degli Abruzzi, Italy) with 20% glycerol, and *Streptococcus mutans* DSM 20523, stored at the same conditions in Trypticase Yeast Extract broth (ISP, Liofilchem, Roseto degli Abruzzi, Italy), were used.

The fungus was subcultured in Sabouraud Dextrose agar (SDA, Liofilchem, Roseto degli Abruzzi, Italy), and, before each assay, one isolated colony was inoculated in SDB and grown aerobically at 37 °C overnight at 170 rpm. *S. mutans* was subcultured in Trypticase Yeast Extract agar (ISP agar, Liofilchem, Roseto degli Abruzzi, Italy), and, prior to the experiments, one isolated colony was inoculated in 10 mL of Yeast Extract broth (ISP, Liofilchem, Roseto degli Abruzzi, Italy) and grown anaerobically at 37 °C overnight. An aliquot of each culture (300 μL) was transferred into a new fresh liquid medium and grown under the same growth conditions until stationary growth phase was achieved.

#### 2.5.2. Antimicrobial Activity of Graphene–PMMA Resin Specimens

Previous to each assay, the graphene–PMMA resin specimens were submitted to UV light for 30 min to avoid contamination. For the bioassay purposes, each microorganism culture, grown overnight and in stationary phase, was diluted tenfold in culture medium to a final concentration of ~10^8^ colony-forming unit per milliliter (CFU/mL) (corresponding to 0.5 Macfarland). The diluted microbial suspensions were equally distributed in sterilized microtubes (1 mL per tube) containing each graphene–PMMA resin specimen. A microbial control without any disk (Ct), to evaluate the growth profile of each microorganism over time, and a resin control (Gc) (nondoped with graphene), to evaluate the antimicrobial effect of PMMA resin, were also included. The suspensions were incubated at 37 °C under a stirring rate of 120 rpm. To evaluate the growth inhibition effect of the specimens, the CFU/mL was determined. Thus, aliquots of samples and controls were collected at times 0 h, 24 h and 48 h of incubation; serially diluted in Phosphate-Buffered Saline (PBS, NZYTech, Lisbon, Portugal); and pour plated in triplicate in SDA for *C. albicans* and ISP agar for *S. mutans*. Prior to serial dilutions after the incubation period, all microtubes were sonicated for 1 min. The Petri plates were incubated at 37 °C for 24 h (plates containing *S. mutans* were incubated at the same temperature in anaerobic chambers), and the viable microbial density was expressed as Log CFU/mL. Three independent experiments, with two replicates (*n* = 2) for each condition, were performed for each assay. The average of the results was calculated.

#### 2.5.3. Surface Adhesion Studies

The microbial cultures of *C. albicans* and *S. mutans* that were subcultured according to the previous conditions were equally distributed in a 96-well microplate (150 μL per well) containing each sterilized graphene–PMMA resin specimen. A resin control (Gc) (nondoped with graphene) was included. The suspensions were incubated at 37 °C under a stirring rate of 120 rpm for 48 h. The samples were fixed in 2.5% glutaraldehyde in PBS for 40 min at RT. After three washes in PBS, the samples were dehydrated through a graded ethanol (VWR Chemicals, Leuven, Belgium) series (15%, 30%, 50%, 70%, 90%, 96%, 3 × 100%, 15 min each) [29,30]. The disks were dried and glued onto stubs, sputter-coated with gold-palladium and examined with a Hitachi S-4100 scanning electron microscope (Hitachi High-Technologies Corp., Tokyo, Japan) operated at 15 kV. The photos show representative spectra of the surface of each disk at different points (*n* = 3).

### 2.6. Statistical Analysis

Statistical significance of surface roughness was determined by Kruskal Wallis analysis of variance with a post hoc test (Bonferroni) using Statistical Package for the Social Sciences (SPSS 25.0, Chicago, IL, USA). A *p* value < 0.05 was indicative of statistical significance.

The statistical analysis of the antimicrobial activity was performed using GraphPad Prism 9.0.0 (GraphPad Software, San Diego, CA, USA). The results were presented as mean of at least 3 independent assays with 2 replicates per condition and the respective mean ± standard deviation. The significance of microbial concentration between graphene–PMMA resin specimens, and along the experiments, was assessed by two-way ANOVA analysis of variance. Tukey’s multiple comparison test was used for a pairwise comparison of the means. The significance of differences was evaluated by comparing the results obtained in the test samples with results obtained for the control (Ct) at different times. A value of *p* < 0.0001 was considered statistically significant.

## 3. Results

### 3.1. Characterization of Graphene–PMMA Resin

The Raman spectroscopy results showed that the graphene platelets consist mainly of few-layer graphene and that photo-polymerized resin is largely PMMA. Raman spectroscopy was performed to identify and evaluate the quality and structural properties of the graphene according to the intensity, frequency and linewidth of its G, D and 2D Raman-modes. The Raman spectra of graphene are characterized by two main peaks: the primary in-plane vibrational mode, G (~1580 cm^−1^, and 2D (~2690 cm^−1^), a second overtone of a different over plane vibration, D (~1350 cm^−1^).

Graphene was detected in every sample, and the introduction of graphene into the resin system did not induce evident changes in the typical graphene Raman spectra (Figure 4).

### 3.2. Surface Roughness

Surface roughness was evaluated (Figure 5), and it was verified to increase with the increase in the graphene concentration, with statistically significant differences in the concentrations of 0.25 (*p* = 0.006) and 0.5 wt% (*p* = 0.005) in comparison with neat resin. In the lower concentration (0.01 wt%), the resin’s surface roughness was approximately the same as neat resin.

### 3.3. Antimicrobial Activity of Graphene–PMMA Resin against C. albicans and S. mutans

The antimicrobial activity of each graphene–PMMA specimen was evaluated against two common microorganisms present in oral biofilms: the yeast *C. albicans* and the Gram-positive bacteria *S. mutans*. The different graphene–PMMA specimens, corresponding to different percentages of graphene, were incubated with each microorganism, and the growth inhibition effect was evaluated after 24 and 48 h of exposure. The results are presented in Figure 6 (a) for *C. albicans* and (b) for *S. mutans*.

The results showed that when *C. albicans* (Figure 6a) was incubated with the control Gc, no fungal growth inhibition was achieved compared to the control, even after 48 h of incubation (ANOVA, *p* = 0.7213). However, the growth inhibition of *C. albicans* was observed when this fungus was incubated with the specimens doped with different percentages of graphene for 24 h (ANOVA, *p* < 0.0001), with no significant recovery after 48 h of incubation. It was also observed that there were no significant differences amongst the graphene–PMMA doped with different percentages of graphene. These results made it clear that not only does graphene exert the growth inhibition of *C. albicans*, but also that the antimicrobial activity effect of the graphene–PMMA does not depend on the graphene amount. 

Regarding *S. mutans*, the antimicrobial effect of the graphene–PMMA specimens had promoted a different profile (Figure 6b). As in the previous case, the control Gc did not inhibit the bacterial growth, even after 48 h of incubation (ANOVA, *p* > 0.9999). In the case of specimens doped with 0.01, 0.1 and 0.25 wt% graphene, this effect was not observed in the first 24 h; however, after 48 h of incubation, the bacterial survival rate decreased in the *c.a.* of 1.5 log CFU/mL when compared to the initial time. In what concerns the specimens containing 0.5 wt% of graphene, the growth inhibition of *S. mutans* was observed after 24 h of incubation (ANOVA, *p* < 0.0001), and after 48 h, the bacterial inactivation of 1.6 log CFU/mL was achieved (ANOVA, *p* < 0.0001). These results suggest that PMMA specimens doped with graphene are capable of inducing the inactivation of *S. mutans* after 48 h of exposure. 

### 3.4. Surface Adhesion Studies 

The surface adhesion of the yeast *C. albicans* and the Gram-positive bacteria *S. mutans* biofilms on different graphene–PMMA specimens with different percentages of graphene was studied by SEM. Thus, each microorganism was incubated in the presence of each specimen for 48 h. This incubation time was chosen according to the antimicrobial activity studies. Figure 7 and Figure 8 show SEM micrographs of the adhesive effect of graphene–PMMA resin specimens against *C. albicans* and *S. mutans* biofilms, respectively.

The results showed that the density of microbial biofilms decreased when the PMMA resin disk (Gc) was compared with the graphene–PMMA resin disks. These results are in accordance with the ones achieved in the microbial growth inhibition studies where the inhibition growth of *C. albicans* and *S. mutans* was more pronounced in the graphene-doped-PMMA resins (Figure 6). 

## 4. Discussion

PMMA is a dental material widely used and is still a clinician’s first choice for several prothesis fabrications. Despite its biocompatibility, reliability, ease handling and low toxicity, this polymer has poor antimicrobial activity which contributes to several oral infections [31,32]. To overcome this disadvantage, several studies have been envisaging the importance of PMMA resins doped with graphene [33]. These new biomaterials take advantage of the important properties of PMMA and of the thermal, mechanical, electrical and antimicrobial activity of graphene [32,34]. In fact, graphene and its derivatives have been considered important tools for the next revolution in dental and medical technology [34,35,36]. Graphene is biocompatible, prevents corrosion and has antimicrobial properties to prevent the colonization of bacteria [37,38,39]. The antimicrobial activity mechanism of graphene has been attributed to its capability to induce oxidative stress, membrane stress and electron-transfer-promoting cell membrane disruption [38]. Moreover, coatings with this material enhance the adhesion of cells and osteogenic differentiation [35].

Herein, 3D-printed PMMA dental resins, reinforced with different concentrations of graphene (0.01, 0.1, 0.25 and 0.5 wt%), were prepared and characterized, and their antimicrobial activity against two common microorganisms present in oral biofilms and their microbial adhesion properties were studied. We had demonstrated that the surface roughness of the graphene–PMMA specimens increased with the increase in graphene concentration. However, despite the increase in surface roughness, graphene proved to have an antimicrobial activity. It was clear that PMMA doped with different concentrations of graphene inhibited the growth of *C. albicans* after incubation for 48 h. Moreover, the graphene–PMMA specimen at a higher concentration (0.5 wt%) had not only inhibited the *S. mutans* growth after 24 h of incubation, but also had inactivated this bacterium after 48 h, showing its bactericide properties. The same bactericide effect was afforded for all the remaining graphene concentrations after 48 h of incubation. The surface adherence studies were in line with these results once the density of microbial biofilms on the graphene–PMMA specimen decreased with the increase in the graphene concentration in the graphene–PMMA resin disks. 

Although the approach for PMMA resins doped with graphene preparation is not the same, some studies regarding their antimicrobial activity and/or the adhesion properties of PMMA–graphene hybrids were reported. Lee et al. [31] reported the incorporation of graphene oxide nanosheets (nGO) into PMMA to introduce sustained antimicrobial-adhesive effects by increasing the hydrophilicity of PMMA. Despite differences in study time, as in our case, the results suggested that PMMA–graphene exhibited better continuous anti-adhesive effects against *C. albicans* after incorporating nGO than those exhibited by pure PMMA via increased hydrophilicity, which envisage the potential usefulness of graphene as a promising antimicrobial dental material for dentures, orthodontic devices and provisional restorative materials. 

Our results suggested that for the inhibition of *C. albicans* and for the inactivation of S. *mutans* after 24 and 48 h of incubation, respectively, there are no significant differences in the graphene concentration of the PMMA–graphene specimen, with a necessary minimum concentration of 0.01 wt% of graphene to exert those effects. In 2019, Said et al. [40] reported the antibacterial effect against *S. mutans* and the thermal expansion of commercially available PMMA incorporated with nano graphene oxide in different concentrations (0, 0.05, 0.1, 0.15 and 1 wt%). In what concerns the antimicrobial effect of these hybrids, 0.5 wt% of graphene oxide was the minimal inhibitory concentration achieved for inhibiting the growth of *S. mutans*. In a similar work, the antimicrobial properties against *E. coli* and *P. aeruginosa* of PMMA fibers containing 2, 4 and 8 wt% of graphene nanoplatelets were studied. [41] The results showed that only PMMA fibers with the highest concentration of graphene had the capacity to inhibit the growth of such bacteria. In a recent study focused on the antifungal activity against the *C. albicans* of acrylic-based tissue conditioner doped with 1, 3 and 5 wt% of nano graphene oxide, the growth inhibition of *C. albicans* was observed only for the highest concentration of graphene oxide [42]. The fact that, for our biomaterial, the minimal concentration to observe antimicrobial effects was lower than the concentrations reported to produce similar materials could be explained by the 3D-printed methodology that was used for the preparation of the graphene–PMMA specimens. In fact, several studies showed the high potentiality of this methodology for the preparation of new polymeric biomaterials with enhanced antimicrobial activity [43,44]. It is worth mentioning that a high degree of dispersion of nano graphene platelets in the resin is required for a successful 3D printing, which in turn is expected to potentiate its antimicrobial activity. The type of PMMA resin and its processing technique can influence the bioavailability of graphene, and this can explain the promising results of our study. 

## 5. Conclusions

This study reported the effect of GNP added to a 3D-printed PMMA resin on surface roughness and the antimicrobial activity against *C. albicans* and *S. mutans*. The results seem to demonstrate an increase in surface roughness of PMMA–graphene disks with the increase in graphene concentration. Additionally, antimicrobial activity against *C. albicans* and *S. mutans* was achieved for the specimens doped with graphene. Surface adhesion studies corroborate that these materials have anti-adhesive effects against the two main microorganisms responsible for prosthetic stomatitis. These findings open the perspective of the use of 3D-printed PMMA–graphene materials in medical and dental applications, namely in denture prosthetics, but future investigations are needed to understand the mechanism of action of PMMA-graphene disks.

## Figures and Tables

**Figure 1 biomedicines-10-02607-f001:**
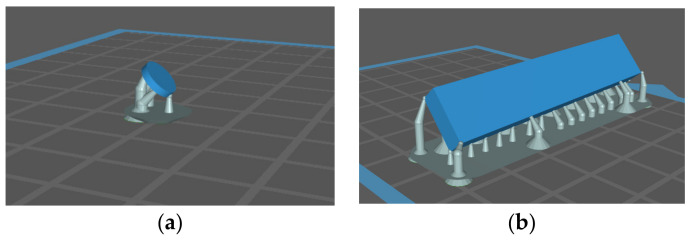
CAD design of specimens used in antimicrobial activity tests (**a**) and in superficial roughness measurement (**b**).

**Figure 2 biomedicines-10-02607-f002:**
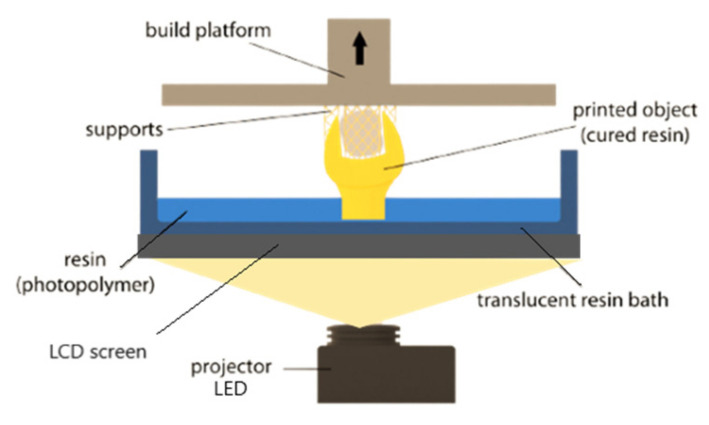
Schematic illustration of an LCD printer fabrication process.

**Figure 3 biomedicines-10-02607-f003:**
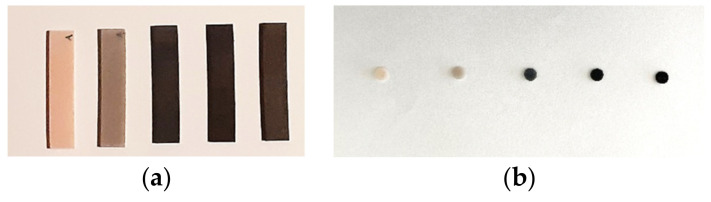
Printed specimens used in surface roughness measurement (**a**) and in antimicrobial activity study (**b**).

**Figure 4 biomedicines-10-02607-f004:**
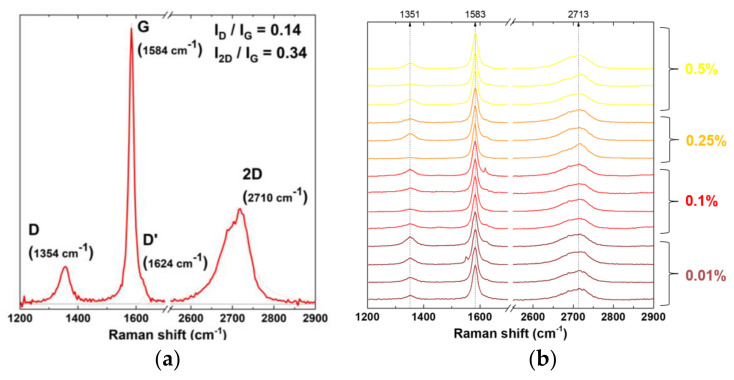
Raman spectroscopy analysis: (**a**) Raman spectra of pristine graphene powder; (**b**) Raman spectra of graphene present in the four concentrations of reinforced resin.

**Figure 5 biomedicines-10-02607-f005:**
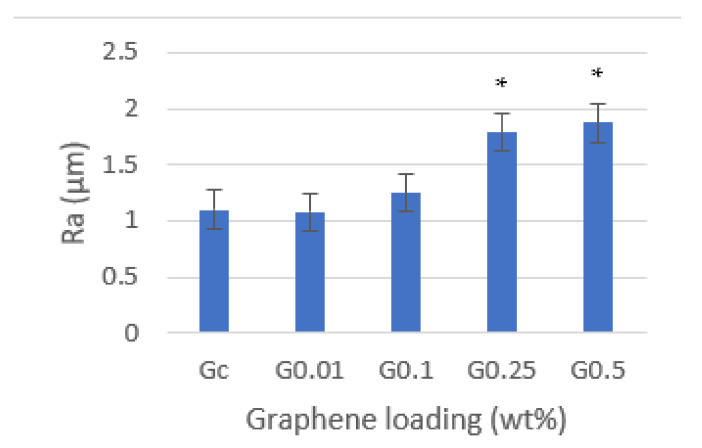
Effect of graphene on PMMA resin’s surface roughness. Mean results (± standard deviation) are presented as Ra (μm) measured using contact profilometer. Asterisks (*) indicate statistical significance compared to Gc (*p* < 0.05).

**Figure 6 biomedicines-10-02607-f006:**
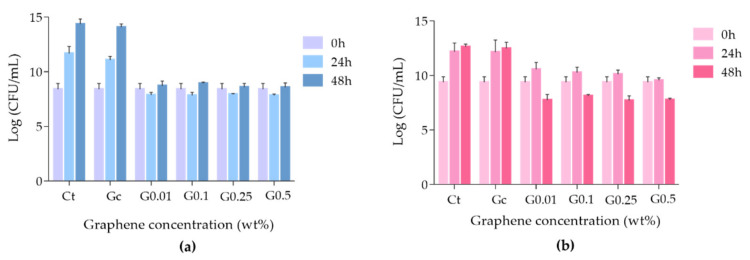
Growth inhibition of *C. albicans* ATCC 11225 (**a**) and *S. mutans* DSM 20523 (**b**) in the presence of the graphene–PMMA resin specimens after 24 and 48 h of incubation. The values are expressed as the means of three independent experiments; error bars indicate the standard deviation.

**Figure 7 biomedicines-10-02607-f007:**
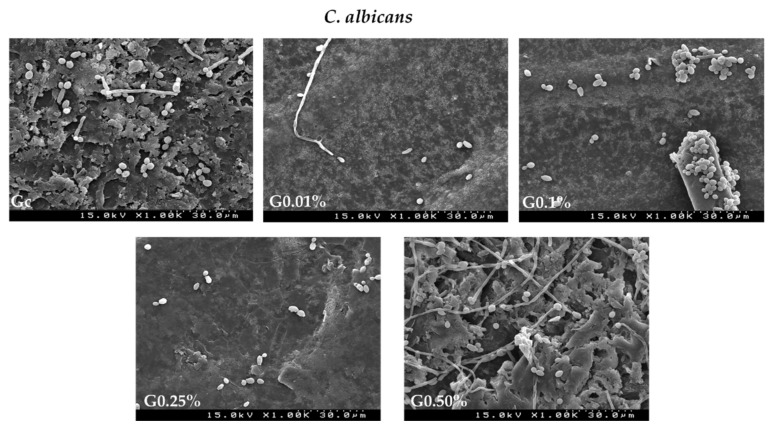
SEM images of graphene–PMMA specimens with different percentages of graphene incubated with *C. albicans* for 48 h. Number of cells visualized—Gc: 59; 0.01 wt%: 12; 0.1 wt%: 47; 0.25 wt%: 26; 0.5 wt%: 30.

**Figure 8 biomedicines-10-02607-f008:**
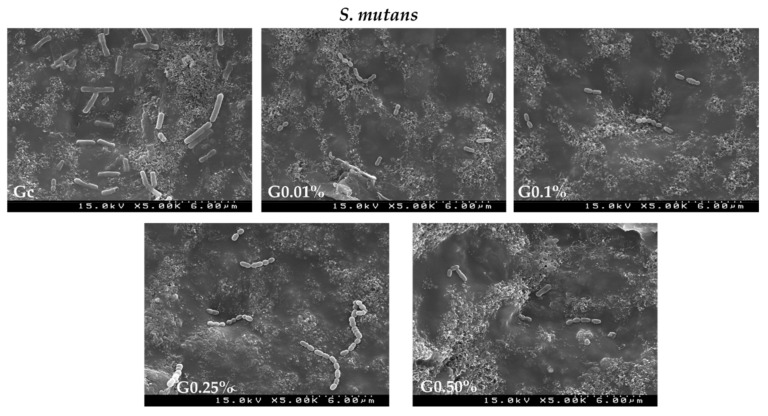
SEM images of graphene–PMMA specimens with different percentages of graphene incubated with *S. mutans* for 48 h. Number of cells visualized—Gc: 34; 0.01 wt%: 17; 0.1 wt%: 14; 0.25 wt%: 18; 0.5 wt%: 11.

## Data Availability

Not applicable.

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
