# Peer review of "Antimicrobial Activity of a 3D-Printed Polymethylmethacrylate Dental Resin Enhanced with Graphene"

_biomedicines, 2022, doi:10.3390/biomedicines10102607_

Round 1

Reviewer 1 Report

In ‘Antimicrobial activity of a 3D printed polymethylmethacrylate dental resin enhanced with graphene’, Salgado et al. tested 3D printed PMMA with different concentrations of graphene and measured the surface roughness as well as C. albicans and S. mutans inhibition of growth and surface adhesion. The surface roughness increased with increasing graphene concentration, but all materials showed some level of antimicrobial activity.

Overall, the manuscript reports some results that could be interesting, but many details about the study are missing or incorrect, as summarized in the comments below. As a result, the conclusions may be over-exaggerated, and additional work to characterize the materials and analyze the obtained results is strongly recommended.

Comments:

-       The authors claim to have used digital light processing (DLP), but they have used a Phrozen Mini 4K printer, which does not use this technology but uses LCD.

-       It would be interesting to actually report the measured viscosities for the resins with the different concentrations of graphene.

-       The method for mixing the graphene into the resin seems to result in the presence of aggregates. Why weren’t techniques, such as ultrasound, applied to better disperse the graphene? This would likely affect the surface roughness.

-       Section 2.5.2 only describes the methods for measurements up to 24 hours, but the results report results up to 48 hours of incubation. The methods should be updated.

-       In Figure 4, the amount of graphene visible on the surface does not seem to correlate with the concentration used. Is the graphene remaining on the surface of the printed structures or is it well distributed throughout the PMMA?

-       It would be helpful to report the actual p-values obtained from the statistical analysis because it is not clear, for example, when p < 0.0001 is not considered significant and p > 0.0001 is considered significant. It would also be helpful to indicate the significant differences in the figures themselves.

-       In section 3.4, the conclusion that the density of the microbial biofilms is decreasing with the increasing concentration of graphene is not well supported by the data. It is not clear if this is referring to the number of yeast/bacteria per area, which could be quantified and reported, or to matrix production in the biofilm.

Author Response

Comments:

-       The authors claim to have used digital light processing (DLP), but they have used a Phrozen Mini 4K printer, which does not use this technology but uses LCD.

We thank the reviewer for alerting us for this mistake. We correct the type of printing technology used.

-       It would be interesting to actually report the measured viscosities for the resins with the different concentrations of graphene.

The viscosity of the resins reinforced with the different graphene concentrations were tested and this property did not suffer significant changes with the presence of graphene, remaining at the value of 1.2Pa.S. This information has been added to the manuscript.

-       The method for mixing the graphene into the resin seems to result in the presence of aggregates. Why weren’t techniques, such as ultrasound, applied to better disperse the graphene? This would likely affect the surface roughness.

We appreciate the suggestion and consider that this aspect may be a limitation of our study. We intend at future investigations to improve this situation by testing other methods of mixing graphene in resin. However, according to Nauser's review (https://doi.org/10.1080/25740881.2018.1563112) pristine graphene is considered to be problematic to disperse in polymer and may aggregate due to high specific surface area, van der Waals, and π–π interactions.

-       Section 2.5.2 only describes the methods for measurements up to 24 hours, but the results report results up to 48 hours of incubation. The methods should be updated.

We thank the reviewer for alerting us for this mistake. We have updated the methods in the revised manuscript.

-       In Figure 4, the amount of graphene visible on the surface does not seem to correlate with the concentration used. Is the graphene remaining on the surface of the printed structures or is it well distributed throughout the PMMA?

We appreciate the comment and consider that figure 4 is not very clear and so it can induce doubts to readers. For this reason, we decided to remove this figure from the manuscript.

 -       It would be helpful to report the actual p-values obtained from the statistical analysis because it is not clear, for example, when p < 0.0001 is not considered significant and p > 0.0001 is considered significant. It would also be helpful to indicate the significant differences in the figures themselves.

We thank the reviewer for this pertinent comment. In fact, the p-value presented in the submitted version is not correct, but in the revised manuscript this typo was corrected and in the case p> 0.0001 (not significant) we presented the exact p-value.

-       In section 3.4, the conclusion that the density of the microbial biofilms is decreasing with the increasing concentration of graphene is not well supported by the data. It is not clear if this is referring to the number of yeast/bacteria per area, which could be quantified and reported, or to matrix production in the biofilm.

We thank the reviewer for alerting of this inconsistency. In the revised manuscript we clarify this point and insert in the figure caption the quantification of yeast/bacteria.  

Reviewer 2 Report

There remain some issues to be addressed before publishing.

1. More experiments should be performed to investigate the mechanism of antimicrobial activity.

2. Photo of the printed specimens should be provided.

3. Can the authors explain more about the Raman spectroscopy? At least the peaks.

4. What are we looking at from Figure 4b to 4d?

5. It is confused why the antimicrobial activity was not enhanced from G0.1 to G0.5? Does it mean that graphene has little effect on the antimicrobial activity? What’s the set of your control group? It may be necessary to provide another control group dopped with different materials.

6. Quantification for Figure 7 and Figure 8 should be performed.

Author Response

  1. More experiments should be performed to investigate the mechanism of antimicrobial activity.

We thank the reviewer for the careful analysis of our work. Although the study mechanism of action of PMMA – graphene specimens was not the focus of our study, and since the biological activity of our specimens only occur in the presence of graphene, we admit that the mechanism of action is due to graphene. Several studies on the graphene antimicrobial activity mechanism have been proposed: oxidative stress, membrane stress, and electron transfer as the main causes of membrane disruption.  We have enriched our discussion in the revised manuscript with this information.

  1. Photo of the printed specimens should be provided.

We appreciate the suggestion, and the image of printed specimens has been added to the manuscript.

  1. Can the authors explain more about the Raman spectroscopy? At least the peaks.

We appreciate the suggestion and add the information regarding the characteristic peaks of the Raman spectrum of graphene.

  1. What are we looking at from Figure 4b to 4d?

We appreciate the comment and consider that figure 4 is not very clear and so it can induce doubts to readers. For this reason, we decided to remove this figure from the manuscript.

  1. It is confused why the antimicrobial activity was not enhanced from G0.1 to G0.5? Does it mean that graphene has little effect on the antimicrobial activity? What’s the set of your control group? It may be necessary to provide another control group dopped with different materials.

We thank the reviewer for the pertinent comment. Our findings corroborate the antimicrobial properties of graphene reported by other research groups. Our results show the antimicrobial effects of graphene, although, in the range of the tested concentrations, it is not possible to observe differences on its activity. We believe that if the graphene concentration was increased, the antimicrobial activity would increase, however more studies are needed to better understand this effect. Nevertheless, our results show that a minimum concentration of 0.01wt% of graphene is necessary to exert antimicrobial effect on C. albicans and S. mutans, lower than the concentrations reported for producing similar materials. In the antimicrobial activity studies, we included two control groups: microbial control without any disk (Ct) to evaluate the growth profile of each microorganism over time and a resin control (Gc) (non dopped with graphene) to evaluate the antimicrobial effect of PMMA resin (this information was highlighted in the revised manuscript). We did not consider other materials since our focus is PMMA – resin dopped with graphene.

  1. Quantification for Figure 7 and Figure 8 should be performed.

As suggested by the reviewer, in the revised manuscript we clarify this point by adding in the figures 7 and 8 caption the quantification of yeast/bacteria.   

Round 2

Reviewer 1 Report

The abstract still refers to digital light processing as the printing technique.

The new text added at the end of section 2.1 is not clearly written.

The schematic in Figure 2 doesn't correctly show the LCD printing technique.

The statement about the p-values considered significant (p<0.0001) is inconsistent with the statistical analysis presented in the text in some places (e.g., p=0.006 and p=0.005 are considered significant and p<0.0001 is not considered significant).

In section 3.4, it is not clear if the results being reported have been observed for replicate samples. Having n=1 is not sufficient to be able to draw conclusions.

Author Response

The abstract still refers to digital light processing as the printing technique.

We thank the reviewer for alerting us for this mistake. We correct the type of printing technology used.

The new text added at the end of section 2.1 is not clearly written.

We apologize for the mistake. The text was incompleted. We add the corret text.

The schematic in Figure 2 doesn't correctly show the LCD printing technique.

We appreciate the comment and change the schematic in figure 2.

The statement about the p-values considered significant (p<0.0001) is inconsistent with the statistical analysis presented in the text in some places (e.g., p=0.006 and p=0.005 are considered significant and p<0.0001 is not considered significant).

We thank the review for alerting us for this inconsistency once again. To avoid misinterpretation, in the revised manuscript the significance of differences was presented only by comparing the results obtained in the test samples with the results obtained for the control (Ct), at different times.

In section 3.4, it is not clear if the results being reported have been observed for replicate samples. Having n=1 is not sufficient to be able to draw conclusions.

We thank the review for the pertinent comment. Although this study was performed in duplicate, our main goal was to observe the surface of each disk at different representative points (n = 3). The photos show representative spectra of sample surfaces.

Reviewer 2 Report

Authors have addressed most of my concerns.

Just one more question, for Figure 7 and Figure 8, why the quantification of cell number is listed as wt%? It is a little confused how the authors performed the quantification.

Author Response

Just one more question, for Figure 7 and Figure 8, why the quantification of cell number is listed as wt%? It is a little confused how the authors performed the quantification.

We thank the reviewer for revising our manuscript. The quantification was made by counting the microorganisms observed in each field and are listed as number of microorganisms per condition. Ex.: in GC were observed 59 cells: Gc: 59; in graphene-PMMA specimen 0.01wt% were observed 12 cells: 0.01wt%:12, etc. It is important to highlight that the photos show representative spectra of sample surfaces.